# Neoadjuvant Intraperitoneal Chemotherapy in Patients with Pseudomyxoma Peritonei—A Novel Treatment Approach

**DOI:** 10.3390/cancers12082212

**Published:** 2020-08-07

**Authors:** Aruna Prabhu, Andreas Brandl, Satoshi Wakama, Shouzou Sako, Haruaki Ishibashi, Akiyoshi Mizumoto, Nobuyuki Takao, Kousuke Noguchi, Shunsuke Motoi, Masumi Ichinose, Yang Liu, Yutaka Yonemura

**Affiliations:** 1Department of Surgical Oncology, Thangam Cancer Center, Namakkal 637001, Tamil Nadu, India; dranuprabhu.surgeon@gmail.com; 2Digestive Unit, Champalimaud Foundation, 1400-038 Lisbon, Portugal; andreas.brandl@fundacaochampalimaud.pt; 3Department of Surgery, Graduate School of medicine, Kyoto University, Kyoto 606-8303, Japan; wakama9@kuhp.kyoto-u.ac.jp; 4Department of Regional Cancer therapy, Peritoneal Surface Malignancy Center, Kishiwada, Tokushukai Hospital, Kishiwada, Osaka 596-0042, Japan; sako-rgb.009@cap.ocn.ne.jp (S.S.); haruaki.ishibashi@tokushukai.jp (H.I.); lymikeleo@hotmail.com (Y.L.); 5NPO to Support Peritoneal Surface Malignancy Treatment, Japanese/Asian School of Peritoneal Surface Oncology, Kyoto 600-8189, Japan; 6Department of Regional Cancer Therapy, Peritoneal Surface Malignancy Center, Kusatsu General Hospital, Shiga 525-8585, Japan; mizumotoakiyoshi1206@yahoo.co.jp (A.M.); nt421500@gmail.com (N.T.); nogupinbad@yahoo.co.jp (K.N.); motoi@kusatsu-gh.or.jp (S.M.); ichinose1967@hotmail.co.jp (M.I.)

**Keywords:** pseudomyxoma peritonei, neoadjuvant intraperitoneal chemotherapy, cytoreductive surgery, hyperthermic intraperitoneal chemotherapy, pathological response

## Abstract

Neoadjuvant intravenous chemotherapy in patients with pseudomyxoma peritonei (PMP) has not shown convincing results. The effectiveness of neoadjuvant intraperitoneal (IP) chemotherapy has never been reported. This prospective, non-randomized phase II study included patients with PMP treated between May 2017 and December 2018, who were not considered suitable for primary cytoreductive surgery (CRS) and hyperthermic intraperitoneal chemotherapy (HIPEC). The majority of patients were treated with laparoscopic HIPEC (oxaliplatin 200 mg/m^2^, 60 min, 43 °C). IP chemotherapy was started 2 weeks after docetaxel 40 mg/m^2^ + cisplatin 40 mg/m^2^, accompanied by oral S1 (tegafur, gimeracil, and oteracil) (50 mg/m^2^) for 14 days, followed by one week rest. Clinical parameters and complications were recorded. In total, 22/27 patients qualified for CRS and HIPEC after neoadjuvant treatment. A complete cytoreduction (Completeness of cytoreduction Score 0/1) could be achieved in 54.5%. The postoperative morbidity rate was 13.6% and mortality was rate 4.5%. In total, 20/22 patients had major pathological tumor responses. The mean drop in CEA was 28.2% and in the peritoneal carcinomatosis index (PCI) was 2.6. Positive or suspicious cytology turned negative in 69.2% of patients. Thus, for PMP patients who were not amenable for primary surgery, the majority received complete cytoreduction after treatment with neoadjuvant IP chemotherapy, with satisfying tumor regression and with low complication rates. The oncological benefit in terms of survival with this new treatment regimen needs to be proven.

## 1. Introduction

Pseudomyxoma peritonei (PMP) is a clinical syndrome wherein the peritoneal cavity gets filled with free or organized mucin, with or without neoplastic cells. In more than 90% of cases, the origin of PMP is a ruptured mucinous appendiceal tumor [1,2]. The tendency of these tumors to remain confined to the peritoneal cavity makes them amenable for treatment with aggressive loco-regional therapies.

### 1.1. Cytoreductive Surgery and Hyperthermic Intraperitoneal Chemotherapy

In the 1990s, before the advent of cytoreductive surgery (CRS) and hyperthermic intraperitoneal chemotherapy (HIPEC), the treatment for PMP mainly comprised repeated drainage of mucin or palliative debulking surgery, with or without palliative chemotherapy. These treatment modalities were associated with a 10-year overall survival (OS) rate of approximately 20% [3,4]. With the introduction of CRS and HIPEC, the survival of these patients has greatly improved to reported 10- year OS rates of 63–74% in selected patients [5,6]. These results have led to CRS and HIPEC being considered the “gold standard” in the treatment of PMP.

Studies have reported prognostic factors such as disease burden, completeness of cytoreduction, and tumor grade as the major factors associated with improved survival following CRS and HIPEC. Youssef et al. reported 5-year and 10-year predicted survival rates of 87% and 74% in patients who had complete cytoreduction as compared to 34% and 23% in patients with major tumor debulking [7]. Ansari et al., from the Basingstoke group, reported that 738/1000 patients with peritoneal malignancy of appendiceal origin after complete cytoreduction had similar 5- and 10-year survival rates of 87.4% and 70.3%, respectively [6].

### 1.2. Neoadjuvant Intraperitoneal Chemotherapy

The use of neoadjuvant intravenous chemotherapy in patients with PMP with high tumor burden as an attempt to improve rates of complete cytoreduction negatively impacted overall and disease-free survival [8]. Nonetheless, the effectiveness of neoadjuvant intraperitoneal (IP) chemotherapy in these patients has never been studied. Yonemura et al. first published the results of neoadjuvant intraperitoneal and systemic chemotherapy (NIPS) in 61 patients with peritoneal carcinomatosis from gastric cancer in 2006 [9]. NIPS contained a combination of IP docetaxel treatment with cisplatin and oral S1 (tegafur, gimeracil, and oteracil). In 22 out of 38 (57.9%) patients who showed positive cytology at the beginning of the treatment, the cytology became negative after NIPS. One-quarter of the patients underwent complete cytoreduction and experienced improved survival. Another study by Fujiwara et al. reported a major response rate of 56% after treatment with NIPS in 25 gastric cancer patients with peritoneal dissemination [10].

Intraperitoneal chemotherapy in neoadjuvant settings has also been used in patients with advanced ovarian cancer. In a pilot study of 10 patients with peritoneal metastasis of ovarian cancer (stage IIIc), Francisco et al. reported decreases in the peritoneal carcinomatosis index (PCI) and CA125 levels after weekly IP paclitaxel treatment in combination with three weekly intravenous carboplatin treatments [11].

Thus, with the effectiveness of neoadjuvant IP chemotherapy becoming gradually evident in cancers such as gastric and ovarian cancers, we decided to evaluate its role in patients with PMP. Therefore, we conducted a phase II study evaluating the effectiveness of neoadjuvant IP chemotherapy in patients with PMP, who were not suitable for primary CRS and HIPEC.

## 2. Results

In total, 27 patients were included in the study and treated according to the therapeutic flowchart (Figure 1). The baseline demographic, clinicopathological, and treatment characteristics of all the patients treated with neoadjuvant IP chemotherapy are shown in Table 1. In total, 22 of 27 (81.5%) patients qualified for CRS and HIPEC after a median of 5 cycles (SD 2.4) of neoadjuvant treatment. One patient was not treated with CRS and HIPEC due to excellent response after 8 cycles of IP chemotherapy, as proven by MRI and decreased tumor marker levels. Laparoscopy, for the purpose of confirming radiological findings and restaging, showed a decrease of the PCI score from 31 (before starting IP chemotherapy) to 8. This asymptomatic patient has been scheduled for observation. The remaining 4 patients were deemed physically unfit for CRS.

The appendix was the primary site of PMP in 85.2% of the patients. Elevation of CEA was noted in 23 of 27 patients, with a mean CEA level of 47.7 (SD 95.0). Neoadjuvant laparoscopic HIPEC (with simultaneous IP port insertion) was performed in 19 of 27 patients (15 of these 19 patients underwent CRS and HIPEC after neoadjuvant IP chemotherapy). In the remaining 8 patients, wherein laparoscopic HIPEC was not feasible due to large omental cakes, IP port insertion was carried out under local anesthesia (7 of these 8 patients similarly underwent CRS and HIPEC after neoadjuvant IP chemotherapy). The mean number of cycles for IP chemotherapy was 5.2 (SD 2.4). Complete cytoreduction (CC0/1) was achieved in 54.5% of the patients (12/22). The postoperative morbidity was 13.6% and mortality was 4.5%. The median length of postoperative stay was 17.5 ± 4.4 days, with no readmission within 90 postoperative days. The tumor regression grades were grade III, grade II, grade I, and grade 0 in 18.2% (4/22), 72.8% (16/22), 9.1% (2/22), and 0% (0/22) or patients, respectively.

The mean drop in CEA was 28.2%. For patients with a CEA > 50 ng/mL (pre-IP chemotherapy), the median decline in CEA was 61.1%, while for patients with CEA < 50 ng/mL (pre-IP chemotherapy), the median decline in CEA was 41.2% (*p* = 0.17) (Figure 2). The decline in CEA correlated indirectly with the tumor regression grade, but failed to reach the significance level (*p* = 0.17) (Figure 3). The relation between pathological tumor regression and histopathological subtype is illustrated in Table 2. The mean decrease in the PCI scores in 15 patients who underwent staging laparoscopy with HIPEC (mean PCI 23.8, SD 10.7) followed by IP chemotherapy and CRS (mean PCI 21.2, SD 14.4) was 2.6. A positive or suspicious cytology became negative in 69.2% of the patients.

No treatment-associated complications were observed during NIPS and no surgical treatment was delayed.

## 3. Discussion

### 3.1. NIPS

IP chemotherapy allows a higher drug concentration intraperitoneally compared to systemic chemotherapy, resulting in better response in terms of peritoneal metastasis, along with less systemic toxicity [12,13]. Yonemura et al., first reported the effectiveness of NIPS in patients with peritoneal metastasis from gastric cancer [9]. Since then, several reports have confirmed the usefulness of NIPS in patients with advanced gastric cancer, with improved survival in patients with pathological response and complete cytoreduction [9,10,14]. Toxicity, especially of grades III, IV, and V, during neoadjuvant IP chemotherapy has also been reported to be lower when compared with systemic chemotherapy [10,15,16].

Our study is the first of its kind to show the effectiveness and feasibility of NIPS in patients suffering PMP who are not amenable for upfront CRS. Our study demonstrated a drop in the mean PCI by 2.6, as well as changes in ascitic fluid cytology from positive or suspicious to negative in 69.2% of the patients, with the possibility of achieving complete cytoreduction (CC0/1) in 54.6% of these patients.

A comparison of the efficacy between NIPS and systemic chemotherapy (SC) proved to be difficult, as the evidence for SC in patients with PMP is limited due to the established standard of care, which is CRS and HIPEC. Some retrospective studies initially revealed a poor or even worse outcome of patients who received preoperative SC compared to patients that received primary CRS [4]. More recent studies using modern chemotherapy, such as mitomycin C and capecitabine or fluorouracil with capecitabine in combination with platinum, showed improvement in 38% of cases, while 55.6% showed benefits or disease control after a median follow up of 24 months [17,18]. The analysis of the largest cohort treated with SC revealed that mainly patients with high grade PMP benefit from perioperative SC and that neoadjuvant SC should be reserved for borderline resectable disease distribution [19]. Correlating the results of the present study with the literature, the efficacy was similar, when comparing rate of complete cytoreduction after neoadjuvant treatment of 54.5% and a CEA drop of 41.2% with average values of <50% and 33–50%, respectively. Regarding the morbidity of the treatment, NIPS seems to be less toxic based on the available information for neoadjuvant SC (grade 3–4 toxicity of 6%). Given the limited level of evidence, it might be interesting to conduct a randomized controlled trial comparing NIPS with neoadjuvant SC for borderline resectable PMP, especially for the high-grade subtype.

### 3.2. Tumor Marker Correlation

Elevated tumor markers (CEA/CA19.9/CA 125) generally correlate with a high disease burden, and patients with PMP and elevated tumor markers consequently reach lower rates of complete cytoreduction [20,21]. In total, 23 out of 27 patients presented with elevated CEA levels. Declines in tumor marker levels after treatment with neoadjuvant chemotherapy have been shown to be associated with better prognosis. Yamao et al. were amongst the first to demonstrate the usefulness of monitoring tumor marker levels in predicting response to chemotherapy [22]. In a study of 26 patients with advanced gastric cancer and with elevation of at least one of the three tumor makers (CEA, CA 19.9, CA 125) before starting systemic chemotherapy, the authors demonstrated a significant difference in survival amongst responders (assessed by drop in tumor markers levels) as compared to non-responders (*p* = 0.0056) [22]. Similarly, Lee et al. demonstrated that the decrease in the tumor marker level after chemotherapy was associated with improved survival in advanced biliary tract cancer patients [23]. They reported tumor marker decline as an independent positive prognostic factor for improved time to progression (TTP) (adjusted HR 0.44; *p* = 0.03) and overall survival (adjusted HR 0.37; *p* = 0.001) in these patients after the first cycle of chemotherapy [23]. In our study, we were able to demonstrate an absolute decline in CEA levels after treatment with neoadjuvant IP chemotherapy, although a survival benefit is yet to be determined. 

### 3.3. Tumor Regression Grading

In a study by Zhang et al., the pretreatment CEA level was identified as one of the strong predictors of tumor regression after treatment with neoadjuvant chemoradiation in patients with locally advanced rectal cancer [24]. Patients with poor tumor regression grades were more likely to have elevated pretreatment CEA levels compared to those with better tumor regression grading (*p* = 0.002) [24]. Our analysis demonstrated that the decline in CEA levels correlated well with our newly implemented tumor regression grading (Table 3), although was not statistically significant owing to the rather small sample size (*p* = 0.17). As neoadjuvant chemotherapy followed by CRS and HIPEC in patients with PMP is a new treatment concept, we had to develop a new tumor regression grading system for these patients. This was created in accordance with the already available tumor regression grading scales existing for different cancers.

There are several reports of tumor regression grading having prognostic significance [25,26,27,28]. Patients with better tumor regression after neoadjuvant treatment are known to have improved disease-free and overall survival rates compared to patients with no tumor regression. We were able to demonstrate a definite decline in CEA levels and improved tumor regression in these PMP patients after treatment with NIPS, with no added morbidity or mortality.

We highlight that tumor regression grade seems to correlate indirectly with the mean CEA drop. The negative correlation is mainly explained by the fact that patients with high-grade PMP express higher levels of CEA compared to patients with acellular mucin or low-grade PMP. In total, four patients with excellent tumor regression at time of CRS and HIPEC treatments were classified as having acellular mucin, without any significant CEA change. On the contrary, patients with high-grade PMP showed significant decreases in CEA, but only 3 of 8 patients (high-grade and high-grade with signet cells) showed pathological regression. This seems to be a paradox, however in fact tumor response can be measured by both decreased CEA and pathologic tumor regression. The performance status of the patient and the distribution of the peritoneal involvement with the ability to reach a complete cytoreduction with an acceptable morbidity and mortality are at least as important in the evaluation of this new therapeutic concept.

### 3.4. Study Limitations

Although we were able to demonstrate a response in patients with advanced PMP after treatment with NIPS (in the form of decrease in CEA levels, tumor regression, decreased PCI, as well as change in ascitic fluid cytology) leading to increasing rates of complete cytoreduction, survival data are currently unavailable to further substantiate the role of NIPS in this group of patients.

## 4. Material and Methods

This prospective, non-randomized exploratory intervention study included patients with PMP from May 2017 to December 2018. Written informed consent was obtained from all patients. This study was carried out in accordance with the Declaration of Helsinki and approval of the ethics committee (H 19-2008) was obtained from Kishiwada Tokushukai hospital. Patients with PMP who were not amenable for upfront CRS (on the basis of CT or MRI PCI scores or raised tumor markers, as per the decision of the treating physician); with good performance status (ECOG 0/1); and who had normal cardiac, renal, liver, and bone marrow functions were selected to undergo this treatment. Radiologic worrisome signs, such as scalloping of organ surfaces (liver or spleen) and the presence of mesenteric foreshortening of the small bowel, were used in order to predict irresectable disease [29]. Patients with previous surgeries with extensive adhesions causing a potentially problematic IP port placement were excluded. Additionally, patients with recurrent PMP were excluded.

### 4.1. Neoadjuvant Laparoscopic HIPEC

The safety and efficacy of neoadjuvant laparoscopic HIPEC in gastric cancer patients has been very well demonstrated by Yonemura et al. [30]. Similarly, neoadjuvant laparoscopic HIPEC was performed in this study. Laparoscopic HIPEC was omitted in patients with large omental cakes. A staging laparoscopy was performed under general anesthesia and the PCI score was documented for the 13 regions of the abdomen and pelvis [31]. The presence or absence of ascitic fluid was documented and ascitic fluid or peritoneal lavage were analyzed by cytological examination. Extensive intraperitoneal lavage (EIPL) was performed with 10 L of saline after draining all the ascitic fluid (if present) [32]. Neoadjuvant laparoscopic HIPEC was then performed with oxaliplatin 200 mg/m^2^ in 4–5 L of saline solution at 43 °C for 60 min. The IP port system was introduced after the completion of laparoscopic HIPEC.

### 4.2. Normothermic Intraperitoneal Chemotherapy

IP chemotherapy was started 2 weeks after the laparoscopic HIPEC. Ascitic fluid with mucinous material (if present) was drained each time prior to IP chemotherapy drug instillation. Cytological analysis of ascitic fluid or washings was also carried out each time. Docetaxel 40 mg/m^2^ along with cisplatin 40 mg/m^2^ were introduced in 500 mL of 0.9% saline and then instilled intraperitoneally through the IP port over one hour on day 1. Ringer lactate (1000 mL) was infused intravenously prior to the IP chemotherapy in order to provide adequate hydration to the patient. Oral tegafur, oteracil, and gimeracil combination (TS1) therapy (50 mg/m^2^) was given for 14 days starting from day 1, followed by 1 week of rest. This entire cycle was repeated every 21 days, with tumor marker evaluation performed prior to each cycle of IP chemotherapy. Patients were treated with 6–8 cycles of IP chemotherapy prior to reassessment for CRS. For patients in whom laparoscopic HIPEC was not feasible due to large omental cakes, IP port insertion was carried out under local anesthesia, the technique of which has already been described in the author’s previous article [9].

### 4.3. CRS and HIPEC

CRS was performed after 5–6 weeks of the last cycle of IP chemotherapy. A CT scan was performed to confirm response or stable disease prior to CRS. CRS was not performed in patients who had either complete response to CT or MRI scan or if patients were deemed unfit for the procedure. Before and after CRS, EIPL was performed in all cases. HIPEC was performed with oxaliplatin 200 mg/m^2^ in 4–5 L of saline via the open coliseum technique for 40 min at 43 °C. Bowel anastomosis, if required, was performed after HIPEC. HIPEC was not performed after CRS in patients with excessive blood loss or inability to maintain adequate urine output or mean arterial pressure. Postoperative complications occurring from the day of operation until patient discharge from the hospital were graded according to the Clavien–Dindo classification [33]. The postoperative major morbidity was defined as grade III b and grade IV complications, respectively.

### 4.4. Response Evaluation

The tumor regression grade (Table 2, Figure 4) was calculated as per the histopathological findings for the tissues obtained during CRS.

A new tumor regression grading system was proposed by Yonemura and colleagues for evaluating response to neoadjuvant chemotherapy in patients with PMP, which is similar to the tumor regression grading system given by Dworak et al. [34]. All tumor regression grades for the included patients were assessed by the same experienced pathologist to avoid any observer bias. Multiple sections from different regions were analyzed and the most common regression score was considered as the final tumor regression grade in a given patient.

### 4.5. Statistical Analysis

All statistical analyses were performed using the IBM SPSS Statistics for Windows, Version 22.0 (Armonk, NY: IBM Corp.) or PRISM 6.0 (GraphPad Software, Inc., La Jolla, CA, USA). Data were provided as means and standard deviations or medians and interquartile ranges (IQRs), and dichotomous variables are represented as percentages. For group comparisons, either Mann–Whitney U-tests, Student’s *t*-tests, or the chi-square tests were performed. All statistical tests were two sided and *p*-values < 0.05 were considered statistically significant.

## 5. Conclusions

This phase II trial showed a low morbidity and mortality associated with the neoadjuvant IP treatment of patients with PMP who were not amenable for upfront surgery. Furthermore, we demonstrated a high rate of complete cytoreduction, along with satisfying tumor regression. By achieving these positive tumor results, we will be able to see in the future whether this treatment leads to oncologic benefit in terms of survival for these patients.

## Figures and Tables

**Figure 1 cancers-12-02212-f001:**
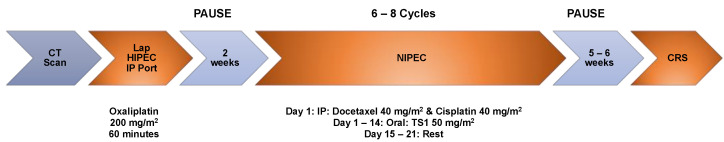
Therapeutic flow-chart of included patients treated with laparoscopic HIPEC, NIPEC, and CRS. CT—computer tomography; Lap—laparoscopic; HIPEC—hyperthermic intraperitoneal chemotherapy; IP—intraperitoneal; NIPEC—normothermic intraperitoneal chemotherapy; CRS—cytoreductive surgery; TS1 (tegafur, gimeracil, and oteracil).

**Figure 2 cancers-12-02212-f002:**
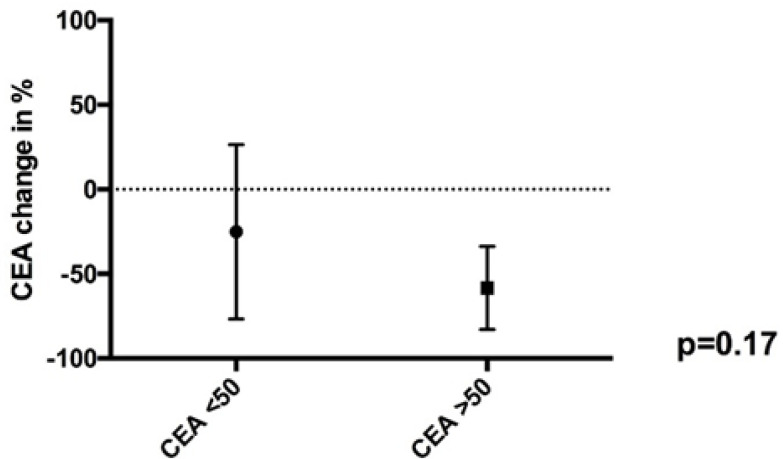
One-way ANOVA data graph for drop in Carcinoembryonic antigen (CEA) as per preintraperitoneal chemotherapy values.

**Figure 3 cancers-12-02212-f003:**
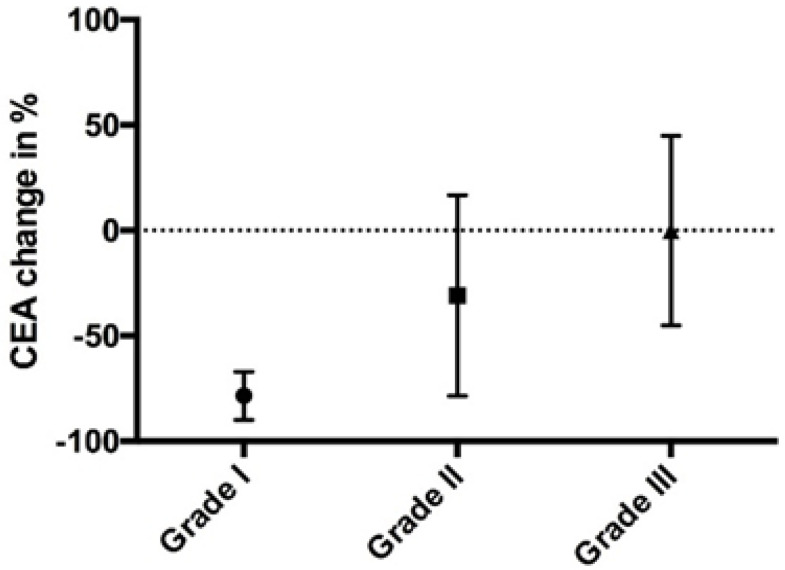
Correlation of tumor regression grade with drop in Carcinoembryonic antigen (CEA) levels (after neoadjuvant intraperitoneal chemotherapy).

**Figure 4 cancers-12-02212-f004:**
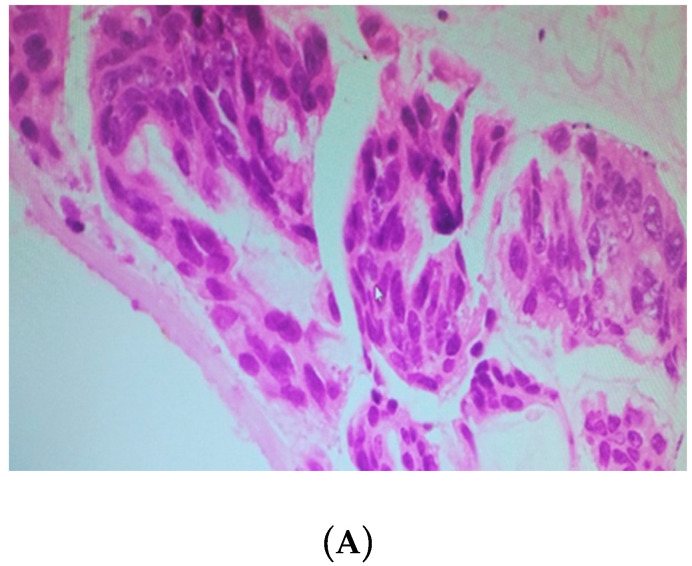
(**A**) Tumor regression grade 0: hematoxylin and eosin (H&E)-stained slide of PMP patient treated with neoadjuvant intraperitoneal chemotherapy, showing viable tumor cells with no response to treatment. (**B**) Tumor regression grade I: H&E-stained slide of PMP patient treated with neoadjuvant intraperitoneal chemotherapy, predominantly showing tumors along with some mucinous material. (**C**) Tumor regression grade II: H&E-stained slide of pseudomyxoma peritonei patient treated with neoadjuvant intraperitoneal chemotherapy, predominantly showing mucinous material with a few scattered tumor cells. (**D**) Tumor regression grade III: H&E-stained slide of pseudomyxoma peritonei patient treated with neoadjuvant intraperitoneal chemotherapy, showing only acellular mucin with no viable tumor cells.

**Table 1 cancers-12-02212-t001:** Baseline demographic, clinicopathological, and treatment characteristics of 27 patients treated with neoadjuvant IP chemotherapy. Continuous variables are demonstrated as means and standard deviations.

Patient Related	
Sex (%)	
Male	37 (10/27)
Female	63 (17/27)
Mean Age [years]	54.6 (SD 12.7)
Mean PCI	22.8 (SD 13.6)
Mean SB PCI	5 (SD 4.7)
**Tumor related**	
**Primary Tumor location (%)**	
Appendix	85.2 (23/27)
Urachus	7.4 (2/27)
Ovary	7.4 (2/27)
**Appearance of PMP (%)**	
Synchronous	74.1 (20/27)
Metachronous	25.9 (7/27)
**Histological type of PMP at CRS (%)**	
Acellular	18.5 (5/27)
Low grade PMP	44.4 (12/27)
High grade PMP	14.8 (4/27)
High grade PMP with signet ring cells	22.2 (6/27)
**Surgery related (*n* = 22)**	
Mean surgery time (min)	314.4 (SD 122.7)
Mean blood loss (ml)	2055.8 (SD 1312.7)
Mean no. of BT units	7.8 (SD 5.8)
**CCR status**	
CC0	40.9 (9/22)
CC1	13.6 (3/22)
CC2	9.1 (2/22)
CC3	36.4 (8/22)
**Clavien–Dindo class (%)**	
Grade 0	68.2 (15/22)
Grade 1–2	0 (0/22)
Grade 3a	13.6 (3/22)
Grade 3b	9.1 (2/22)
Grade 4	4.5 (1/22)
Grade 5	4.5 (1/22)
**Surgical complications (%)**	31.8 (7/22)
Anastomotic leak	13.6 (3/22)
Intra-abdominal abscess	4.5 (1/22)
Pancreatic fistula	4.5 (1/22)
Small bowel perforation	9.1 (2/22)
**Medical complications (%)**	13.6 (3/22)
Pulmonary	4.5 (1/22)
Deep venous thrombosis	4.5 (1/22)
Sepsis	4.5 (1/22)
**Treatment related**	
**Tumor regression grade (%)**	
Grade III	18.2 (4/22)
Grade II	72.8 (16/22)
Grade I	9.1 (2/22)
Grade 0	0 (0/22)

IP—intraperitoneal; PCI—peritoneal carcinomatosis index; SB—small bowel; PMP—pseudomyxoma peritonei; CRS—cytoreductive surgery; BT—blood transfusion; CC: completeness of cytoreduction.

**Table 2 cancers-12-02212-t002:** Tumor regression depending on underlying disease.

Histopathologic Subtype	Tumor Regression (Grade II and III)	No Tumor Regression (Grade 0 and I)
Acellular mucin	4	0
Low-grade	1	7
High-grade	0	2
High-grade with signet cells	2	4

**Table 3 cancers-12-02212-t003:** Tumor regression grading system after neoadjuvant treatment in patients of pseudomyxoma peritonei.

Tumor Regression Grade	Pathological Finding (Figure 1a–d)
Grade 0	No regression
Grade I	Predominantly tumors, along with mucinous material
Grade II	Predominantly mucinous material, with few scattered tumor cells
Grade III	No viable tumor cells detectable (only acellular mucin)

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
