# Peer review of "Neoadjuvant Intraperitoneal Chemotherapy in Patients with Pseudomyxoma Peritonei—A Novel Treatment Approach"

_cancers, 2020, doi:10.3390/cancers12082212_

Round 1
Reviewer 1 Report
The paper is really interesting and innovative. Results can lead to a new work-up of the patient with unresectable PMP. The statistical analysis is strong and well conducted.
Few comments:
1- Material and Methods should be explained above the discussion
2- It would be easier to follow the whole paper with a scheme presentation of the study setting.
3- In 36% and 12 % of CRS ended respectively with CC3 and CC2. Are dìthere any correlation between CC, PCI and Histotype? It would be interesting to know.
Author Response
Reviewer 1:
The paper is really interesting and innovative. Results can lead to a new work-up of the patient with unresectable PMP. The statistical analysis is strong and well conducted.
Few comments:
1- Material and Methods should be explained above the discussion
RE: The order of the Research Manuscript Sections is defined by the Journal itself. We prepared our manuscript according to the published Instructions for Authors, meaning the Material and Methods section is located between Discussion and Conclusion section.
2- It would be easier to follow the whole paper with a scheme presentation of the study setting.
RE: Thank you for your important comment. We created a therapeutic flow-chart of the included patients and integrated it in the manuscript as new Figure 1.
3- In 36% and 12 % of CRS ended respectively with CC3 and CC2. Are there any correlation between CC, PCI and Histotype? It would be interesting to know.
RE: We explored this aspect and did not find any correlation. Table show number of patients p=0.13
|
|
CC 0/1 |
CC 2/3 |
|
Acellular |
3 |
1 |
|
Low-grade |
4 |
4 |
|
High-grade |
0 |
4 |
|
High-grade with Signet cells |
4 |
2 |
Reviewer 2 Report
The authors submit their prospective single arm phase II study of neoadjuvant HIPEC with oxaliplatin for pseuodmyxoma peritonei. The authors are well experienced in HIPEC and have applied a concept of NIPS which they have studied in other more agressive malignancies and now have applied it to pseuodomyxoma peritonei. This is an important concept as there are many patients who present with unresectable pseudomyxoma peritonei
I have the following comments and suggestions for improvement.
1) As the authors discuss pseudomyxoma peritonei is a clinical term with a heterogenous biology comprising a variety of primary tumor sites. This is both a strength (generalizability) but also a major limitation as the biology of a well-differentiated mucinous adenocarcinoma or LAMN is very different from a mucinous adenocarcinoma with signet ring cells. I think this warrants further discussion. Were the responders (histology and or CEA) more likely to be low grade or high grade?
2) unresectability is a very subjective inclusion criteria. Did the authors use any scoring system besides PCI such as the simplified preoperative assessment for appendix tumor (SPAAT) score (Dineen et al. Annals of Surgical Oncology. 2015 Oct;22(11):3640-6)? This could be applied retrospectively. If not having a figure with a pre-op CT scan showing the extent of disease and then CT scan post NIPS would be very informative.
3) A PCI drop of only 2.6 in a patient with a average PCI of 22.8 does not seem clinically significant. Any factors associated with a robust response?
4) The reported morbidity was for the CRS and HIPEC. Any morbidity/adverse events from the NIPS? IP port infections, clogging or other complications? Any treatment delays?
5) Since the reported morbidity is in-hospital, and not the standard 30-day or 90 day interval the authors should include the re-admission rate and median length of stay.
6) in the methods, the abbreviation EIPL is not spelled out.
7) I think the discussion needs to be expanded. Future directions needs to be discussed. Comparison of neoadjuvant systemic to NIPS therapy for PMP? randomized trial of NIPS or not for borderline resectable PMP?
8) Could the authors expand on why oxaliplatin and not mitomycin C.
Overall great study and look forward to authors comments and improvements.
Author Response
Reviewer 2:
The authors submit their prospective single arm phase II study of neoadjuvant HIPEC with oxaliplatin for pseuodmyxoma peritonei. The authors are well experienced in HIPEC and have applied a concept of NIPS which they have studied in other more agressive malignancies and now have applied it to pseuodomyxoma peritonei. This is an important concept as there are many patients who present with unresectable pseudomyxoma peritonei
I have the following comments and suggestions for improvement.
1) As the authors discuss pseudomyxoma peritonei is a clinical term with a heterogenous biology comprising a variety of primary tumor sites. This is both a strength (generalizability) but also a major limitation as the biology of a well-differentiated mucinous adenocarcinoma or LAMN is very different from a mucinous adenocarcinoma with signet ring cells. I think this warrants further discussion. Were the responders (histology and or CEA) more likely to be low grade or high grade?
RE: Thank you for your important comment. We conducted an additional analysis of our data and created a new Table 2 in order to illustrate the distribution of tumor regression between different pathological subtypes. In addition, we corrected one typographic error regarding the “negative correlation of tumor regression grading and CEA drop”. The negative correlation is mainly explained by the fact, that 4 patients with excellent tumor regression at time of CRS & HIPEC were classified with acellular mucin, without any significant CEA change. On the contrary, patients with high-grade PMP had significant CEA drop, but only 3 of 8 patients (high-grade & high-grade with signet cells) showed pathological regression. This seems paradox, but in fact tumor response can not only be measured in CEA drop, or pathologic tumor regression. The performance status of the patient, as well as the distribution of the peritoneal involvement with the ability to reach a complete cytoreduction are at least as important in the evaluation of this new therapeutic concept. Unfortunately, we are not able to deliver a systematic objective evaluation of the increase of performance status, as you could also argue that a laparoscopy with HIPEC upfront is already improving the performance status, as it evacuates the whole mucin of the abdominal cavity in these patients. Therefore, we believe the combination of tumor regression, rate of complete cytoreduction, and acceptable morbidity/mortality in CRS & HIPEC in these patients is proof enough to support our new concept for these patients, who do not have a lot of alternatives. Further studies including these factors, supporting this hypothesis are on the way, and will provide more results in the future.
We added these aspects to the discussion section.
2) unresectability is a very subjective inclusion criteria. Did the authors use any scoring system besides PCI such as the simplified preoperative assessment for appendix tumor (SPAAT) score (Dineen et al. Annals of Surgical Oncology. 2015 Oct;22(11):3640-6)? This could be applied retrospectively. If not having a figure with a pre-op CT scan showing the extent of disease and then CT scan post NIPS would be very informative.
RE: Thank you for your valuable comment. Indeed, the SPAAT score is a very useful tool in order to predict resectability of PMP, though it was not systematically used in our study. The radiologic evaluation in our study included the presence of worrisome signs, such as scalloping of organ surface (liver or spleen) and the presence of mesenteric foreshortening of the small bowel as described by Dineen et al. Clinical worrisome signs were mainly associated to the performance status, which was significantly affected by patients with far progressed disease. We added these details of the applied radiological criteria to the Methods section.
Due to the provided deadline of 7 days to revise this manuscript, we were unfortunately not able to provide the SPAAT Score for the included patients, as suggested.
3) A PCI drop of only 2.6 in a patient with a average PCI of 22.8 does not seem clinically significant. Any factors associated with a robust response?
RE: The study evaluated CEA decrease, peritoneal cytology, and pathologic regression as objective or robust response parameters during therapy. CEA dropped in median 41.2%, peritoneal positive or suspicious cytology became negative in 69.2%, and 90.1% demonstrated pathologic regression (Grade III and II) after neoadjuvant therapy.
4) The reported morbidity was for the CRS and HIPEC. Any morbidity/adverse events from the NIPS? IP port infections, clogging or other complications? Any treatment delays?
RE: Thank you for your important comment. There were no treatment associated morbidity or adverse events from NIPS in this cohort of patients. We added this information to the results section.
5) Since the reported morbidity is in-hospital, and not the standard 30-day or 90 day interval the authors should include the re-admission rate and median length of stay.
RE: We added this missing detail to the Results section: The median length of postoperative stay was 17.5 ± 4.4 days, with no readmission within 90 postoperative days.
6) in the methods, the abbreviation EIPL is not spelled out.
RE: The abbreviation EIPL appears twice in our manuscript. The first time it is correctly spelled out as “Extensive intraperitoneal lavage (EIPL)” (line 198/199), and the second time the abbreviation was used “Before and after CRS, EIPL was performed in all cases “ (line 220).
7) I think the discussion needs to be expanded. Future directions needs to be discussed. Comparison of neoadjuvant systemic to NIPS therapy for PMP? randomized trial of NIPS or not for borderline resectable PMP?
RE: Thank you for your valuable comment. We enriched the discussion section with a paragraph comparing NIPS with neoadjuvant systemic chemotherapy, as well as the comment for the need of a randomized trial.
8) Could the authors expand on why oxaliplatin and not mitomycin C.
RE: The compound oxaliplatin was chosen, as it is known to be effective in the setting of HIPEC for colorectal cancer. It has also proven systemic efficacy (Level 1) against metastatic high-grade neoplasm or adenocarcinoma of the appendix, as this was the estimated majority of included patients at the time of creating the study protocol.
In the time of negative PRODIGE-7 trial, and the common awareness of the lack of profound basic science data established from animal models of HIPEC protocols and cytotoxic agents, we believe that discussing this aspect does not add value to the manuscript.